# Anticonvulsant Action of GluN2A-Preferring Antagonist PEAQX in Developing Rats

**DOI:** 10.3390/pharmaceutics13030415

**Published:** 2021-03-19

**Authors:** Pavel Mares, Grygoriy Tsenov, Hana Kubova

**Affiliations:** Department of Developmental Epileptology, Institute of Physiology, Czech Academy of Sciences, 14220 Prague, Czech Republic; grygoriy.tsenov@fgu.cas.cz

**Keywords:** NMDA receptors, GluN2A subunit, anticonvulsant action, pentylenetetrazol-induced seizures, cortical epileptic afterdischarges, immature rats

## Abstract

The GluN2A subunit of N-methyl-D-aspartate (NMDA) receptors becomes dominant during postnatal development, overgrowing the originally dominant GluN2B subunit. The aim of our study was to show changes of anticonvulsant action of the GluN2A subunit-preferring antagonist during postnatal development of rats. Possible anticonvulsant action of GluN2A-preferring antagonist of NMDA receptors P = [[[(1S)-1-(4-bromophenyl)ethyl]amino](1,2,3,4-tetrahydro-2,3-dioxo-5-quinoxalinyl)methyl]phosphonic acid tetrasodium salt (PEAQX) (5, 10, 20 mg/kg s.c.) was tested in 12-, 18-, and 25-day-old rats in three models of convulsive seizures. Pentylenetetrazol-induced generalized seizures with a loss of righting reflexes generated in the brainstem were suppressed in all three age groups in a dose-dependent manner. Minimal clonic seizures with preserved righting ability exhibited only moderately prolonged latency after the highest dose of PEAQX. Anticonvulsant action of all three doses of PEAQX against cortical epileptic afterdischarges (generated in the forebrain) was found in the 25-day-old animals. The highest dose (20 mg/kg) was efficient also in the two younger groups, which might be due to lower specificity of PEAQX and its partial affinity to the GluN2B subunit. Our results are in agreement with the postero-anterior maturation gradient of subunit composition of NMDA receptors (i.e., an increase of GluN2A representation). In spite of the lower selectivity of PEAQX, our data demonstrate, for the first time, developmental differences in comparison with an antagonist of NMDA receptors with a dominant GluN2B subunit.

## 1. Introduction

Epilepsy is a common disorder leading to staggering costs, both with regard to human suffering and to economics. There is no causal cure for epilepsy, and the management of epileptic seizures in patients with epilepsy relies mostly on long-term administration of antiepileptic drugs (AEDs). Because epilepsy frequently begins during childhood and has a higher incidence during the first decade of life than at any other time [1], issues of safety and efficacy of AEDs represent a major concern in pediatrics. Despite this, new AEDs are designed, regardless of the specific needs of the developing brain, and few attempts have been made to develop age-specific therapies. Some antiepileptic drugs affecting glutamate ionotropic receptors are used in human medicine—e.g., felbamate (among other mechanisms of action antagonist of N-methyl-D-aspartate (NMDA) receptors [2]) and perampanel—α-amino-3-hydroxy-5-methyl-4-isoxazolepropionic acid (AMPA) receptor antagonist [3].

It is widely accepted that glutamate-mediated hyperexcitation plays a causative role in seizure generation and that suppression of glutamatergic excitation is very effective against seizures experimentally induced in laboratory animals.

Effects of glutamate in synaptic transmission are mediated by specific receptors. Among them, ionotropic NMDA and AMPA receptors represent major targets for preclinical and clinical research in epilepsy and development of new AEDs. Experimental studies have demonstrated that agonists of both NMDA and AMPA receptors elicit seizures in laboratory animals of all ages, while antagonists of these receptors exhibit pronounced anticonvulsant effects [4]. Strong anticonvulsant effects of NMDA receptor antagonists were for the first time demonstrated more than thirty years ago [5]. However, because NMDA receptors are involved in many critical brain functions, it is not surprising that a blockade of NMDA receptors may produce a variety of deleterious neurobehavioral effects [6]. One possible way to overcome this limitation is targeting of particular subtypes or subpopulation of NMDA receptors.

NMDA receptors are widely expressed throughout the central nervous system; their number, localization, and subunit composition differ in a cell- and synapse-specific manner, and subunit expression is strictly regulated during development. NMDA receptors are assembled as tetramers composed of two GluN1 subunits and two GluN2 (A–D) or less common GluN3 (A and B) subunits. GluN2A and GluN2B predominate in the brain. Early in development, expression of GluN1 and GluN2A subunits is low and increases during maturation, whereas GluN2B is highly expressed in the first two weeks of life and decreases thereafter [7,8,9]. The subunit composition determines not only physiological properties of NMDA receptors and their role in pathology of neural disorders but also heterogeneity in pharmacological properties [10,11].

A non-competitive antagonist of NMDA receptors MK 801 (dizocilpine) possesses marked anticonvulsant effects. They have been demonstrated in several models of epileptic seizures across all age groups of laboratory rodents [12,13]. Administration of dizocilpine is, however, associated with many severe adverse effects, such as impairment of cognitive functions and motor performance or neuronal death. Interestingly, a more recently developed blocker of NMDA receptors, memantine, a low-affinity and GluN2A/GluN2B non-specific NMDA antagonist, also exhibits anticonvulsant properties but with significantly reduced side effects [14]. In a model of pentylenetetrazol-induced seizures, memantine was effective in all age groups of developing rats, whereas ifenprodil, a GluN2B subunit-selective antagonist of NMDA receptors, was effective only at early stages of postnatal development [15,16]. The decrease of anticonvulsant efficacy of ifenprodil with age corresponds with the switch of GluN2B and GluN2A subunits during postnatal development. Thus, the anticonvulsant action of GluN2A subunit-preferring antagonists should be expected later in life.

Unfortunately, the selectivity of GluN2A-preferring antagonists that are available for in vivo use is relatively low, and in spite of active research in this field, more selective antagonists described up to now exhibit serious problems. The solubility of TN-201 in physiological saline is poor [17], and MPX-004 and MPX-007, highly selective GluN2A antagonists in tissue slices and cultures, cannot be used for systemic administration [18]. So far, only racemate PEAQX and its stereoisomer NVP-AAM007 are commercially available for in vivo administration. Their selectivity for the GluN2A vs GluN2B subunit is however relatively low—approximately seven times for PEAQX [19] and nine times for NVP-AAM007 [20].

To study the developmental profile of anticonvulsant properties of GluN1/GluN2A-preferring antagonist, we used non-competitive antagonist PEAQX. Brain circuitries that generate various seizure types mature at different developmental rates. Therefore, three models of epileptic seizures with different sites of origin were studied: (1) pentylenetetrazol-induced generalized tonic-clonic seizures originating primarily in the brainstem [21,22], (2) predominantly clonic seizures (“minimal seizures”) induced by pentylenetetrazol originating in the forebrain [21,22] and (3) cortical epileptic afterdischarges characterized by spike-and-wave rhythm in the EEG generated in the cortico-thalamo-cortical circuits [23].

## 2. Materials and Methods

### 2.1. Animals

Experiments were performed in male Wistar albino rats (*n* = 203, Institute of Physiology of the Czech Academy of Sciences) on three postnatal days (P 12, 18, and 25). The day of birth was defined as day 0 and animals were weaned at P21. Rats were housed in a controlled environment (temperature 22 ± 1 °C, humidity 50–60%, lights on 6 am–6 pm) with free access to food and water. During experiments with pups, temperature in Plexiglas cages was maintained at 32 ± 2 °C using an electric heating pad connected to a digital thermometer to compensate for immature thermoregulatory functioning at this age [24].

All procedures involving animals and their care were conducted according to the ARRIVE guidelines (https://www.nc3rs.org.uk/arrive-guidelines) in compliance with national (Act No 246/1992 Coll.) and international laws and policies (EU Directive 2010/63/EU for animal experiments and the National Institutes of Health guide for the care and use of Laboratory animals NIH Publications No. 8023, revised 1978). The experimental protocol was approved 4 June 2018 by the Ethical Committee of the Czech Academy of Sciences (Approval No. 15/2018).

### 2.2. Drugs

PEAQX (P = [[[(1S)-1-(4-bromophenyl)ethyl]amino](1,2,3,4-tetrahydro-2,3-dioxo-5-quinoxalinyl)methyl]phosphonic acid tetrasodium salt; Tocris Bioscience, Abingdon, UK **#** 5018/10) was dissolved in distilled water in concentration of 5 mg/mL and administered in doses of 5, 10, and 20 mg/kg s.c. Doses were selected according to previous studies by others [22]. In all three age groups in both experiments, animals in the control groups received physiological saline in the volume corresponding to the highest dose of PEAQX (i.e., 4 mL/kg s.c.). Pentylenetetrazol (PTZ, Sigma-Aldrich, Dorset, UK; # P6500), freshly dissolved in distilled water (50 mg/mL), was injected 30 min later in a dose of 100 mg/kg s.c.

### 2.3. PTZ-Induced Seizures

A total of 98 animals were used in this experiment (P 12 (*n* = 32), 18 (*n* = 40), and 25 (*n* = 34)). At the dose of 100 mg/kg used in our study, PTZ elicits two types of convulsive seizures, which differ by developmental profile, seizure generator, and by pharmacological sensitivity. Appearance of minimal (mS, mostly clonic convulsions involving head and forelimb muscles with preserved righting reflexes) seizures is age-dependent. They are regularly elicited by PTZ in P15 and older animals but not in younger rats [25]. In contrast, generalized tonic-clonic seizures (GTCS), starting with short running phase and accompanied by a loss of righting reflexes, can be induced already in P1 rats [26].

After PTZ injection, animals were individually placed in Plexiglass cages and their behavior was monitored for 30 min by an experienced observer. Incidence and latency of both seizure types (mS and GTCS) and other behavioral phenomena (e.g., isolated myoclonic jerks) and abnormalities were also recorded. To assess the severity of epileptic phenomena, animals were assigned a score for the most severe behavioral characteristics as follows [27]:0-no changes0.5-abnormal behavior (e.g., automatisms, increased orienting reaction)1-isolated myoclonic jerks2-atypical or prolonged minimal seizures3-clonic seizures (mS) involving head and forelimb muscles with preserved righting reflexes (older term minimal metrazol seizures)4-generalized seizures without the tonic phase (GCS)5-complete generalized tonic-clonic seizures (GTCS)

### 2.4. Cortical Epileptic Afterdischarges

Surgery was performed at P12 (*n* = 31), P18 (*n* = 37), and P25 (*n* = 37) under ether anesthesia. Two stimulation electrodes were placed epidurally over the right sensorimotor cortex (*AP* = −1 and +1, *L* = 2 mm relative to bregma) and registration electrodes were implanted at the corresponding coordinates (*AP* = 0, *L* = 2 mm) in the contralateral hemisphere. Both indifferent and ground electrodes were inserted into the occipital bone. All electrodes were fixed to the skull by means of dental acrylic. After cessation of anesthesia, the animals were allowed to recover for at least 1 h. Then, their reflexes (righting, placing, and suckling) were examined, they were fed with 5% sucrose, and the recording was started. Rat pups were used only once.

EEG experiments were recorded using a custom-made 4-channel preamplifier (Prague, Czech Republic) and CED Power 1401 data acquisition interface with the input range ±5 V. Sample frequency was set on 2000 Hz/channel and band-pass filtering from 1 to 300 Hz. Recordings were analyzed offline using Spike2 (CED, United Kingdom), Brainstorm, and LabChart Reader (AD Instruments, Colorado Springs, CO, USA) software. Signal was first notch-filtered (50 Hz), and then, digital band-pass filter from 1 to 45 Hz was applied. Power spectral analysis was performed from the first 3 s of afterdischarge with the FTT size 1024, Hann (Cosin–Bell) data window without overlapping and the zero-frequency component was removed. In this study, we calculated total power for the first and third ADs in P12, P18, and P25 animals with a PEAQX dose of 10 mg/kg.

Cortical afterdischarges (ADs) were elicited by rhythmic electrical stimulation (15-s series of 1-ms biphasic pulses at 8-Hz frequency) by means of a stimulator with a constant current output. The suprathreshold current intensity necessary for elicitation of an epileptic AD was used throughout the experiment. Intensities were chosen according to our previous data (6 mA in P12 and 3 mA in P18 and P25 animals) [28] to elicit ADs of at least 5-s duration. Stimulation was repeated six times with intervals of 10 min between the end of an AD and subsequent stimulation. PEAQX was administered subcutaneously at the same doses as were used in a model of PTZ-induced seizures (i.e., 5, 10, and 20 mg/kg s.c.). 5 min after the end of the first AD, i.e., subsequent stimulations were applied 5, 15, 25, 35, and 45 min after PEAQX injection.

The EEG activity was recorded before stimulation, during stimulation, and 2 min after the end of the AD. Duration of the ADs was measured and original data were used for statistics. Intensity of movements accompanying stimulation and of clonic seizures accompanying ADs were quantified according to modified Racine’s scale [29].

### 2.5. Statistics

Sample size was determined in advance according to previous experience with the given models and followed the principles of the three R’s (replacement, reduction and refinement; https://www.nc3rs.org.uk/the-3rs). Outcome measures and statistical tests were prospectively selected. At the beginning of the study, a simple randomization was used to assign each animal to a particular treatment group. Data acquisition and analysis were done blinded to the treatment. Data were analyzed using GraphPad Prism 8 (GraphPad Software, San Diego, CA, USA) software. Using the D’Agostino–Pearson normality test, all data sets were first analyzed to determine whether the values were derived from a Gaussian distribution. Outliers were identified with the ROUT test (Q = 1%). Differences in anticonvulsant activity between controls and PEAQX-treated animals were analyzed using ordinary one-way ANOVA with post-hoc multiple comparison by controlling the false discovery rate (FDR). The incidence of individual seizure phenomena was compared first with χ^2^-test for trends, and subsequently control and individual dose groups, using the Fisher exact test. Two-way repeated measure (RM) ANOVA with one between-group factor (control, treatment, or age) and one within-subject factor (repeated session), corrected for multiple comparison by controlling the FDR, was used to compare duration of epileptic afterdischarges between treatment groups within one age group and/or between control groups of various ages. Power spectra from the first and third afterdischarges were analyzed using multiple t-test with FDR. *p*-value < 0.05 was required for significance.

## 3. Results

### 3.1. Pentylenetetrazol-Induced Seizures 

Incidence of generalized tonic-clonic seizures (GTCS) was 100% in controls of all three age groups, i.e., the score expressing seizure severity was 5.0 (Figure 1, Figure 2 and Figure 3). The effects of PEAQX on incidence of GTCS depended on age. Whereas it was suppressed by dose-dependent manner in P25, in both P12 and P18 rats, administration of PEAQX resulted in specific suppression of the tonic phase of GTCS, changing them into gener-alized clonic seizures (GCS); the relation to the dose of PEAQX was outlined. The 20-mg/kg dose significantly decreased incidence of generalized seizures. In compliance with changes in incidence of GTCS, administration of all tested doses of PEAQX re-sulted in a decrease of seizure severity expressed as a score in all age groups (one-way ANOVA *F*_3,28_ = 10.202, *p* < 0.001; *F*_3,28_ = 19.133, *p* < 0.001; *F*_3,28_ = 6.054, *p* = 0.003 for the three age groups). Because PEAQX in a dose of 20 mg/kg almost completely suppressed incidence of generalized seizures (GS for GTCS and GCS together), this dose was ex-cluded from statistical analysis of latencies of GS. In two lower doses, 5 and 10 mg/kg, PEAQX tended to prolong latencies of GS in P12 and P25 animals (*F*_2,19_ = 2.607, *p* = 0,1 and *F*_2,15_ = 1.106, *p* = 0.357, respectively), this prolongation was significant in P18 ani-mals (*F*_2,18_ = 8.516, *p* = 0.002).

Minimal clonic seizures (mS) with preserved righting ability were seen in the ma-jority of P18 and P25 controls (Figure 2 and Figure 3). The incidence of these seizures was not sig-nificantly affected by any dose of PEAQX. PEAQX tended to increase latencies of mS in either age group (one-way RM ANOVA—*F*_3,26_ = 1.853, *p* = 0.162 and *F*_3,26_ = 2.05, *p* = 0.131 for 18- and 25-day-old rats, respectively; Figure 2 and Figure 3). The longer latencies after the highest dose of 20 mg/kg seen in both P18 and P25 groups did not reach the level of statistical significance due to high variability of individual values.

### 3.2. Cortical Epileptic Afterdischarges

Control groups: two-way ANOVA demonstrated significant effect of number of AD (*F*_3.153,104.0_ = 28.37; *p* < 0.0001), of age (*F*_2,33_ = 14.77; *p* < 0.0001), and significant interaction (*F*_10,165_ = 4.317; *p* < 0.0001). Multiple comparisons revealed that the third to sixth ADs in 12-day-old rats were longer than corresponding ADs in either older group (Figure 2).

The bottom part of the figure shows examples of original EEG recordings of the third afterdischarges (ADs) in the three age groups (P12, P18, and P25 from top to the bottom). Note the difference in EEG pattern between P12 rats and older animals (right bottom part of Figure 4). There are no spikes in 12-day-old rat pups; AD is formed by sharp waves. Examples represent third second of the third ADs in rats injected with the 10-mg/kg dose of PEAQX (in red frame). Time and amplitude scales are on the bottom.

PEAXQ administration markedly affected duration of ADs. This effect was most marked in 25-day-old animals (Figure 5).

In P12 rats, two-way ANOVA revealed significant effect of number of AD (*F*_2.354,63.55_ = 19.20; *p* < 0.0001), treatment (*F*_3,27_ = 7.154; *p* = 0.0011), and interaction (*F*_15,135_ = 3.302; *p* = 0.0001). Multiple comparisons with control demonstrated that third AD in the 10-mg/kg group was longer than control AD and fourth to sixth ADs in the 20-mg/kg group were shorter than corresponding control ADs.

In P18 animals, according to two-way ANOVA, number of AD was not significant, but treatment (*F*_3,33_ = 3.013; *p* = 0.0439), and interaction (*F*_15,165_ = 2.124; *p* = 0.0111) were. Multiple comparisons indicated shorter fifth and sixth ADs in the 20-mg/kg group than in controls, and fifth AD in the 5-mg/kg group was longer than the control.

Mixed effect analysis was performed on P25 animals instead of two-way ANOVA because one value was missing. Again, the number of AD was not significant but treatment (*F*_3,33_ = 7.498; *p* = 0.0006) and interaction (*F*_15,164_ = 2.690; *p* = 0.0011) demonstrated significant differences. Multiple comparisons with controls revealed that second AD was shorter in 5- and 10-mg/kg groups, the third AD in the 10-mg/kg rats and the fourth to sixth ADs in all three PEAQX groups compared to control.

Cortical stimulation and afterdischarges elicited clonic movements (seizures) of forelimbs (Racine stage 3). An exception was found in one or two animals 18 and 25 days old, where stage four (forelimb clonus + rearing) was observed. Average stage never exceeded the value 3.25 ± 0.164; ANOVA did not indicate significant differences.

Computer analysis of the first three seconds of ADs of the first (i.e., control, pre-drug) and the third ADs (i.e., 15 min after PEAQX administration) demonstrated an increase of total power in 12-day-old rats (significant in the third second of AD) and practically no changes in 18-day-old rats and biphasic change in amplitude—an increase in the fastest component and a decrease in frequencies from 10 to 20 Hz in 25-day-old animals (Figure 6).

## 4. Discussion

The results of the present study demonstrate age-dependent differences in anticonvulsant effects of GluN1/GluN2A-preferring antagonist of NMDA receptors PEAQX. PEAQX exhibited anticonvulsant activity in all age groups of juvenile animals tested, but its effects differ qualitatively, as well as quantitatively, depending on seizure model and age of animals.

It is indeed very difficult to correlate the data obtained in immature rodents with those in humans [30]. Using DNA to estimate cell numbers and cholesterol levels to estimate myelination, Dobbing [31] defined a period of “growth spurt”. In humans this period occurs between the last few weeks of gestation and the first few months of life and corresponds with that of rats aged P10–12. This developmental parallel is supported by similarity in development of pattern of cortical electrical activity in human babies and immature rats [32,33,34]. Although somewhat vague, it is still used for comparison of early developmental stages of human and rodent brain. Sexual maturation in rats takes place between P35–P45 in both genders [35]. According to these data we can presume that rats 12, 18 and 25 days old correspond to infants or early school children, respectively.

The three seizure types used in our study are generated in different brain structures with different courses of maturation. PTZ-induced generalized tonic-clonic seizures are primarily generated in the brainstem [21,22] and spike-and-wave discharges characteristic for CxAD represent a corticothalamocortical phenomenon [23,36]. Minimal clonic seizures present after PTZ administration since the third postnatal week in rats (probably generated in basal forebrain [21,22]) were not markedly affected by PEAQX. Repeating Browning experiments in immature rats, we found that animals with transection of the brainstem were not able to generate minimal clonic seizures. Generalized tonic-clonic seizures were regularly elicited in the transected animals. Spinal cord transection at the low thoracic level did not block the possibility to evoke tonic-clonic seizures even in hindlimbs, only the threshold was higher than for the same phenomenon in forelimbs [22].

PEAQX exhibited anticonvulsant action against PTZ-induced generalized seizures in all three studied age groups. Effects against GTCS are in agreement with an early increase in brainstem expression of the GluN2A subunit, which becomes dominant at postnatal day nine [37]. The preferential suppression of the tonic phase of generalized seizures brings additional support to a possibility that the tonic and clonic phases of generalized seizures have different generators [38,39]. These two phases are probably not tightly connected in younger age groups, as was demonstrated in our previous studies [40]. Unfortunately, the site of origin of minimal clonic seizures is only vaguely localized in the basal forebrain [21] and no data for the time of switch between GluN2A and GluN2B in this area are accessible.

PEAQX was markedly more effective against epileptic cortical ADs in 25-day-old rats compared to younger age groups. Interestingly, PEAQX primarily affected prolongation of AD duration observed with repetitive stimulation. This phenomenon was detected in all age groups used in our study and reflects delayed postictal potentiation clearly expressed 10 min after the electrically induced cortical Ads, e.g., [41]. This postictal potentiation was described in the immature brain in both cortical [42] and limbic [43] structures. Mechanisms responsible for this phenomenon are not fully understood, but our unpublished work (Mareš et al.—in preparation [44]) suggest a dominant role of ionotropic glutamate receptors (both AMPA and NMDA). The effect of PEAQX in 25-day-old rats is in agreement with the later development of the cortical and thalamic GluN2A/ GluN2B ratio (during the third postnatal week [45,46]) and with data demonstrating the increase of cortical and hippocampal expression of the GluN2A subunit at later stages of postnatal development [46]. In infantile rats, only higher doses of PEAQX exhibited partial anticonvulsant effect. In contrast, selective GluN2B antagonists ifenprodil and Ro 25-6981 exhibited a marked anticonvulsant effect in 12-day-old rat pups and a failure of these antagonists in 25-day-old animals in the same model [41,47]. The age-specific effect of ifenprodil was observed, not only in models of epileptic seizures but also in spontaneous behavior in open field—locomotion was significantly increased in 12-day-old rats but not in older animals [16]. Effects of PEAQX on spontaneous locomotion in developing animals remains to be tested. Partial suppression of ADs observed in 12- and 18-day-old rats after the highest dose of PEAQX might be explained by a low selectivity of PEAQX and its effects on receptors containing the GluN2B subunit [17]. The unexpected proconvulsant action of PEAQX in the cortical afterdischarges in the youngest (10-mg/kg dose), and especially in the 18-day-old (5-mg/kg dose) group, has to be further analyzed. Taking our data together, higher efficacy of PEAQX against GTCS matches up to postero–anterior gradient of maturation of forebrain and hindbrain GluN2A subunit of NMDA receptors [37,46].

Several factors, however, hinder the interpretation of results of our study. As already mentioned above, similarly to other GluN2A antagonists available for in vivo administration, selectivity of PEAQX to GluN1/GluN2A receptors vs GluN1/GluN2B receptors is relatively low. Low selectivity resulting in combined antagonistic effects on both GluN1/GluN2A and GluN1/GluN2B are likely involved in anticonvulsant effects of higher doses of PEAQX. Several studies have demonstrated that activation of GluN2A and GluN2B subunits can play different roles in numerous pathophysiological and physiological processes. The underlying mechanisms for this are not yet fully understood, but the differences between the signaling pathways associated with GluN2A and GluN2 likely represent one of them. Some of these signaling cascades are known to participate in regulation of brain excitability [47], and their role in anticonvulsant effects of GluN1/GluN2A vs GluN1/GluN2B receptor antagonists has to be further studied.

In spite of these limitations, our study brings new information on the anticonvulsant activity of GluN2A-preferring antagonist PEAQX, which are different from that of GluN2B antagonists [15,40,47], especially in the developmental profile, and adds data to different effects of GluN2A and GluN2B in the brain [48,49].

## 5. Conclusions

To our best knowledge, this is the first study demonstrating age-related differences in the anticonvulsant effects of GluN1/GluN2A-preferring antagonist PEAQX. We demonstrated that these effects differ according to the age of animals, used dose, and seizure type. In the future, it is crucial to develop highly selective antagonists for GluN1/2A or GluN1/2A/2B [49] receptors to determine suitability of these substances in treatment of childhood epilepsies. They should be tested in specific models of age-related seizures and special attention has to also be paid to the safety of these drugs for the developing brain.

## Figures and Tables

**Figure 1 pharmaceutics-13-00415-f001:**
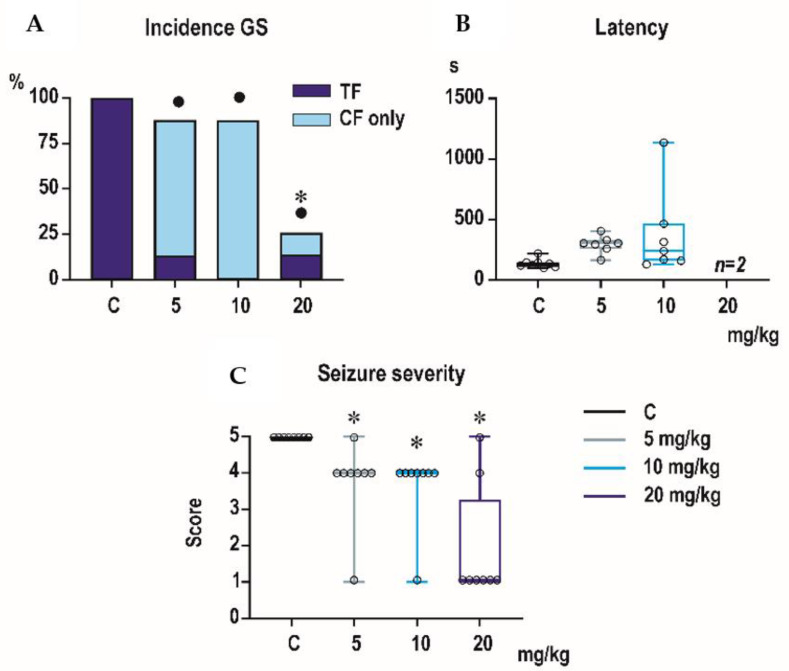
Effect of P = [[[(1S)-1-(4-bromophenyl)ethyl]amino](1,2,3,4-tetrahydro-2,3-dioxo-5-quinoxalinyl)methyl]phosphonic acid tetrasodium salt (PEAQX) on seizures elicited by pentylenetetrazole in postnatal (P)12 pups (each experimental group consists of eight animals). PEAQX in all tested doses affected incidence (expressed in percentage) of generalized seizures (GS) (**A**). In lower doses 5 and 10 mg/kg, PEAQX preferentially suppressed tonic phase of GS, without affecting incidence of clonic phase. The highest dose of 20 mg/kg significantly suppressed the incidence of GS, i.e., both tonic and clinic phase. The percentage of animals exhibiting complete generalized tonic-clonic seizures (dark blue) and generalized seizures without tonic phase, i.e., generalized clonic seizures (light blue) (axis y). Dots mean significant difference in the presence of complete tonic-clonic seizures, asterisk significant difference in the incidence of GS (both generalized tonic-clonic seizures (GTCS) and generalized clonic seizures (GCS) together) in comparison with controls. X axis in all three graphs presents control and three groups with different doses of PEAQX. Administration of PEAQX in doses of 5 and 10 mg/kg resulted in significant prolongation of latencies of GS (latencies in seconds are on y axis) compared to controls. *n* = 2 means that only two animals exhibited generalized seizures (**B**). The highest tested dose of PEAQX significantly decreased seizure severity expressed as score (on axis y; from 1—abnormal behavior but no convulsions to 5—generalized tonic-clonic seizures) (**C**). Data are shown as box plots (min to max) with individual values (circles). Asterisks denote significant difference in comparison with controls.

**Figure 2 pharmaceutics-13-00415-f002:**
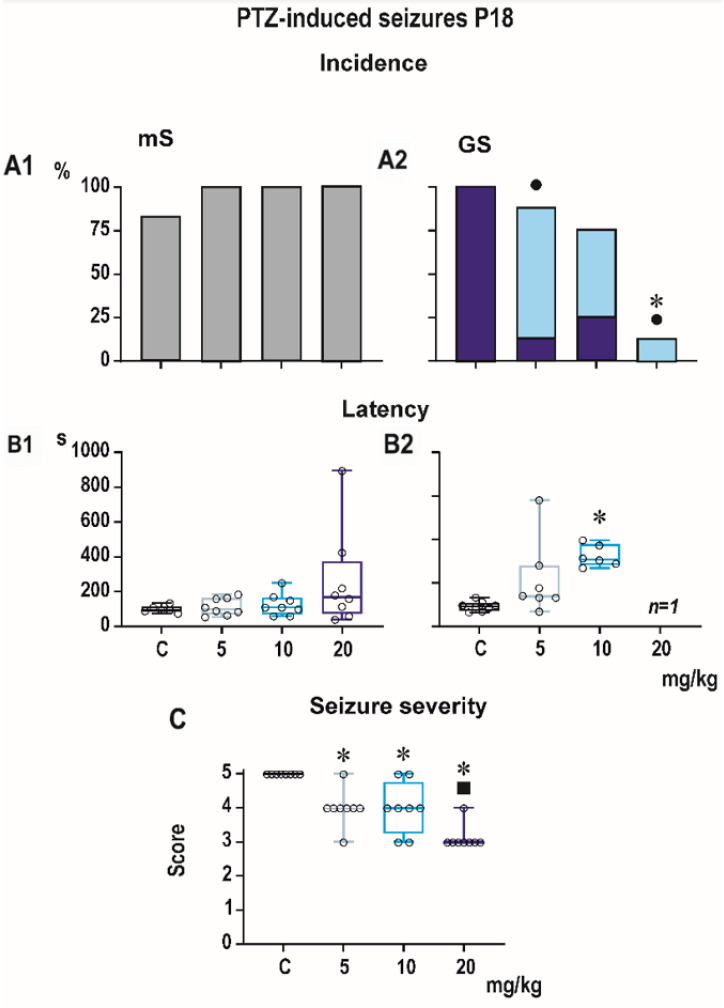
Effect of PEAQX on seizures elicited by pentylenetetrazole in P18 rats (each group consisted of 10 animals). Administration of PEAQX in any tested doses did not affect incidence (**A1**) or latency (**B1**) of minimal seizures (mS). Incidence of tonic phase of generalized seizures was suppressed by PEAQX in a dose of 5 mg/kg, and the highest dose of 20 mg/kg abolished GS in almost all animals in the group (**A2**). Latency to GS was significantly longer in animals receiving PEAQX in a dose of 10 mg/kg (**B2**). Only one rat with the dose of 20 mg/kg exhibited GS (*n* = 1). PEAQX in all three tested doses decreased seizure severity (**C**). Dots mean significantly lower incidence of complete generalized tonic-clonic seizures in comparison with controls. Asterisks denote significant differences in comparison with the controls.

**Figure 3 pharmaceutics-13-00415-f003:**
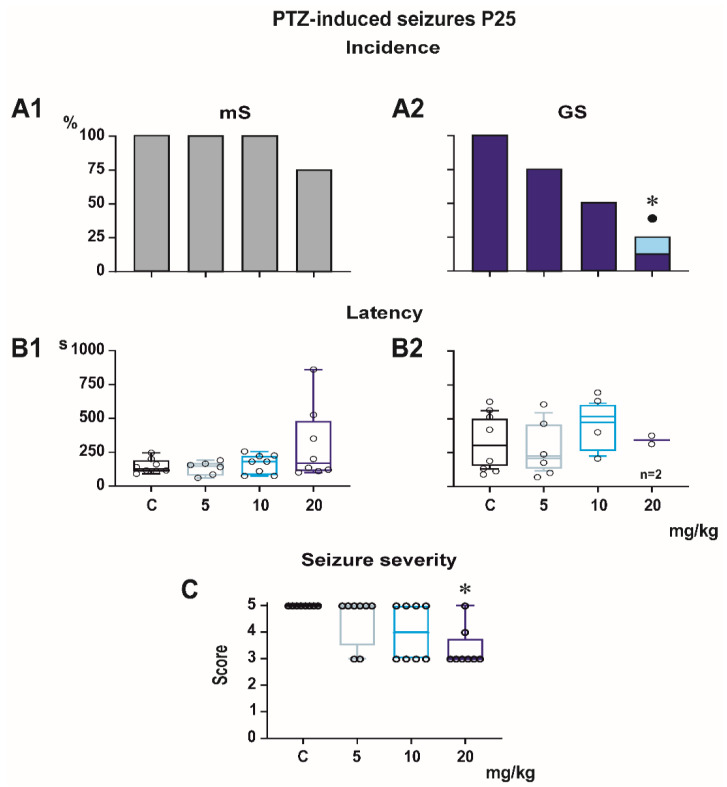
Effect of PEAQX on seizures elicited by pentylenetetrazole in P25 rats (each group consisted of eight animals). Similarly to P18 animals, PEAQX was ineffective against mS (**A1**,**B1**). Only in the highest dose of 20 mg/kg did PEAQX significantly decrease incidence of the tonic phase and GS as a whole (**A2**) and seizure severity (**C**) but not the latency (**B2**). Dots mean significantly lower incidence of complete generalized tonic-clonic seizures in comparison with controls. Asterisks denote significant decrease of complete generalized tonic-clonic seizures (**A2**) and seizure severity (**C**) in comparison with the controls.

**Figure 4 pharmaceutics-13-00415-f004:**
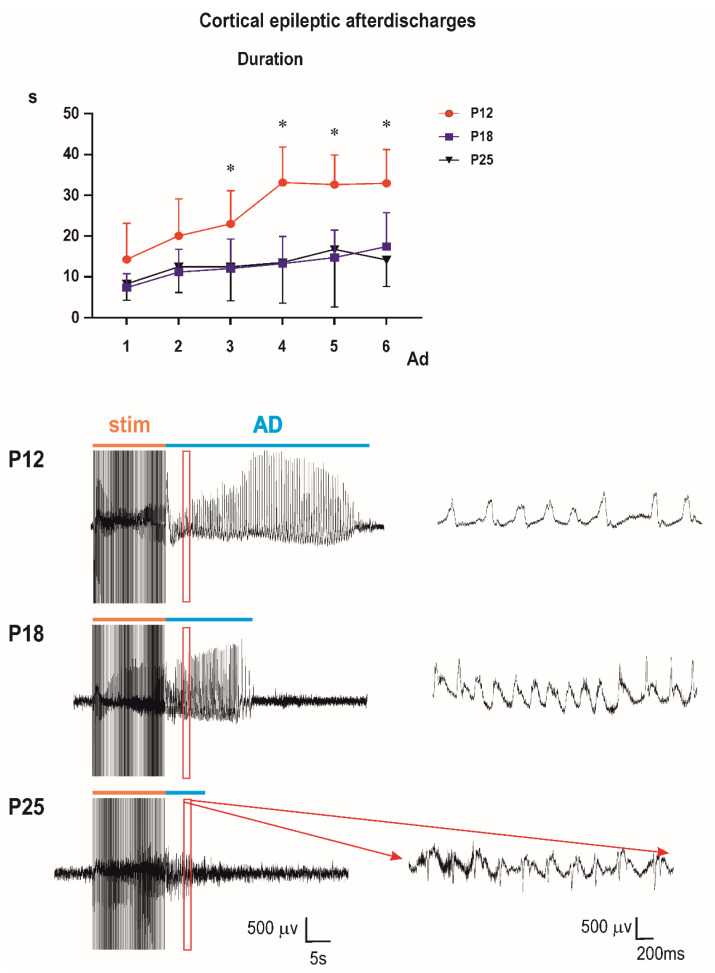
Age-related differences in duration of epileptic cortical afterdischarges (ADs) in control animals. Since the third stimulation, duration of ADs was significantly longer in P12 animals compared to P18 and P25 rats. Data are presented as a mean ± SD (upper part of the figure). * Mean significant difference of values in P12 from corresponding ADs in the other two age groups.

**Figure 5 pharmaceutics-13-00415-f005:**
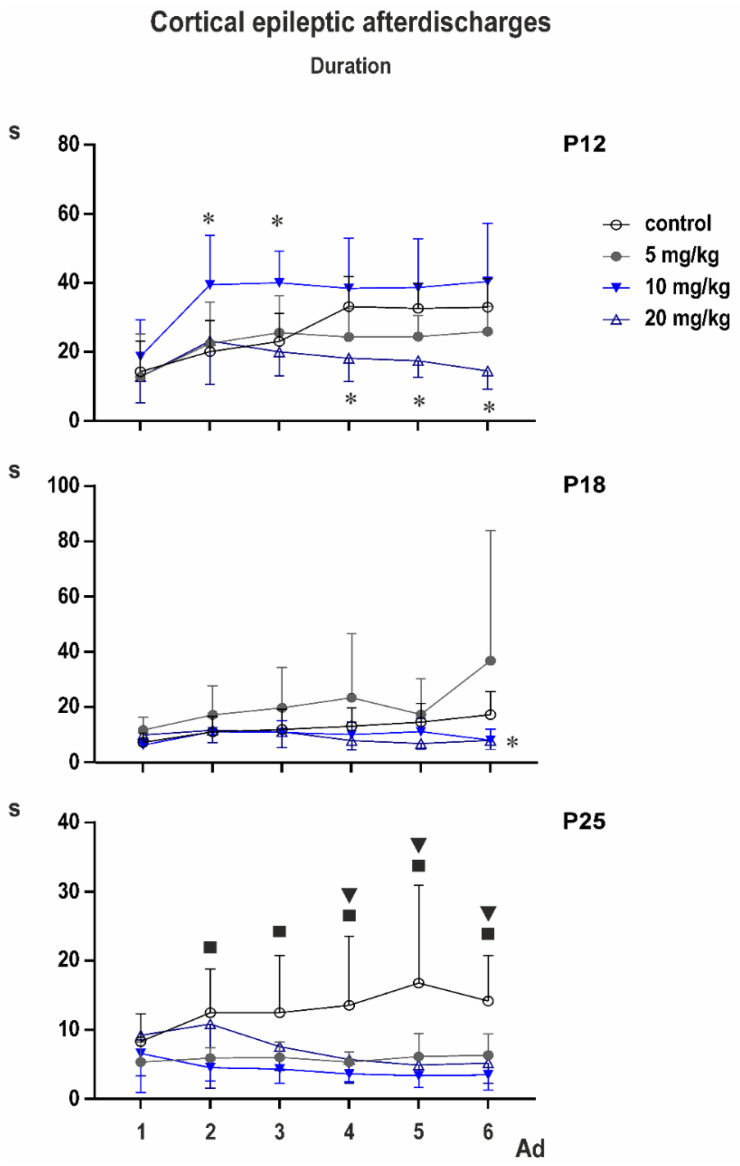
Effect of PEAQX on average duration of epileptic cortical afterdischarges in the three age groups (from top to bottom 12-, 18-, and 25-day-old animals). Stimulation of suprathreshold intensity was repeated six times in 10-min intervals (x-axis—first to sixth AD). PEAQX was injected 5 min after the end of the first AD, and effects of three doses (5, 10, or 20 mg/kg—symbols are in inset) on duration of ADs (in seconds, y-axes) was evaluated. Controls received physiological saline in volume corresponding to the highest dose of PEAQX. * Denotes significant difference from corresponding AD in the control group, square—significant difference between the control and 5-mg/kg groups, triangles—significant difference of controls from the 10-mg/kg group.

**Figure 6 pharmaceutics-13-00415-f006:**
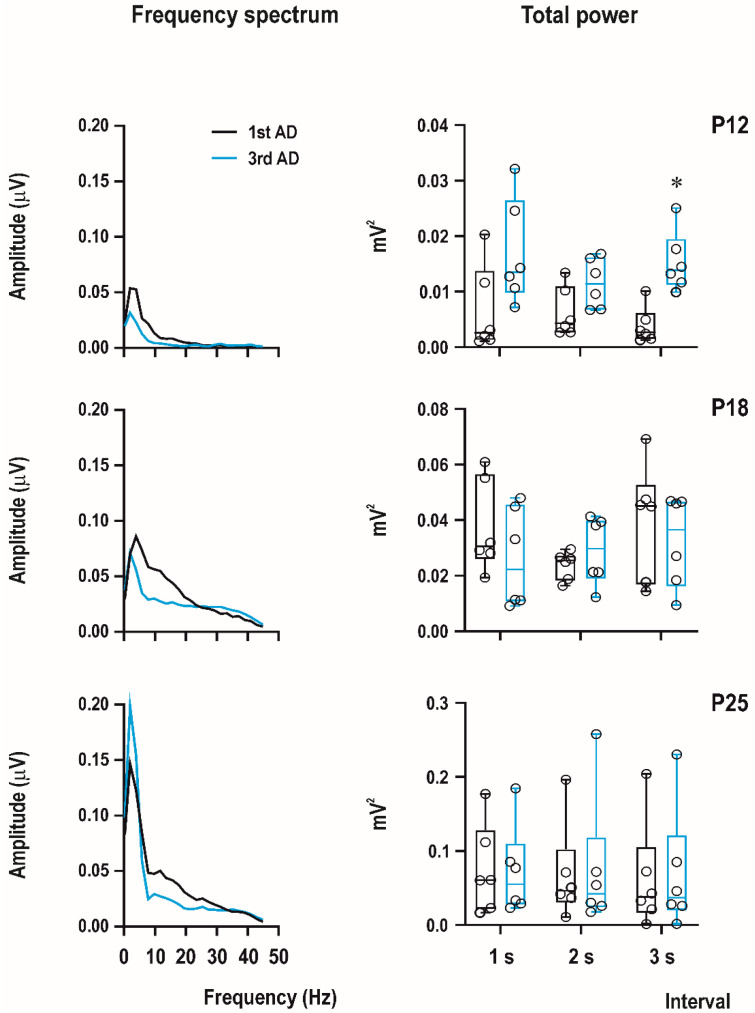
An example of computer analysis of EEG recordings. Comparison of the first three seconds of the first, pre-drug AD (black lines) and the third, post-drug AD (blue lines). PEAQX (10 mg/kg) was administered 5 min after the end of the first AD. Left part of the figure shows frequency spectrum (fast Fourier transform analysis) from all age groups used (from top to bottom P12, P18, and P25). X axis—frequency of EEG waves (Hz), y-axis—amplitude in µV. Right part of the figure illustrates total power in the first three seconds of ADs (x-axis). Y-axis—power in mV^2^. Data are shown as box plots (min to max) with individual values (circles). Asterisk denotes significant difference in total power between interval-matched control and PEAQX groups.

## Data Availability

Data are available in the Institute of Physiology, Czech Academy of Sciences, Prague, Czech Republic.

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
