# Peer review of "Anticonvulsant Action of GluN2A-Preferring Antagonist PEAQX in Developing Rats"

_pharmaceutics, 2021, doi:10.3390/pharmaceutics13030415_

Round 1

Reviewer 1 Report

This work analyzed the potential anticonvulsant action of the GluN2A subunit-preferring antagonist (PEAQX) during postnatal development in 12-, 18- and 25-day-old rats in three models of convulsive seizures. Authors sustain that data demonstrate a developmental difference in comparison with an antagonist of NMDA receptors with a dominant GluN2B subunit.

A major concern is that additional experiments are required before publication. And authors should indicate very clear about the number of animals used in each group.

In this model you have animals of three different ages, three different doses of PEAQX and five stimulations at different times. So you have at least 45 sets of data some independent some not. Please clarify very well experimental design

For all Figures and data presentation: Please use box plots and scatter diagrams for all your graphs. The use of bars plus SD is no longer acceptable for scientific report. See Calin-Jageman RJ, Cumming G (2019) Estimation for better inference in neuroscience.

-----------------

rows 33-36 Please include specific citations and use an example of glutamate related drugs used for treating epilepsy in humans not animal models. I thought this statement is not well sustained.

rows 115-1227. When was the PEAQX administered in this model? How many animals received each dose of PEAQX. Please be more precise in this respect. Please also clarify if experiments were doble blind. There were no control group? Please made a better experimental design description.

Figure 1. Please add clear explicit label to Y axes.

Figure 2. A much more explicit representation of data is required. Also here use box plots and scatter graphs together to better represent your data no longer use of simple bars and deviation.

Please also in all groups be very clear about the number of animals used and inform clearly if number of animals are not enough, it is obvious, that additional experiments are required before publication.

Figure 3. Number of animals in each group. Be explicit with meaning of axis labels. Use box plot and scatter graphs-  Please specify why controls are significant. They are being compared with first measurement ? If so, then you have a time effect?

245-250. This first statement is not sustained by the results.  No intergroup comparisons were shown. All statistical analyses are against control, or it is not clearly documented in the text. Please correct this.

Author Response

Reviewer 1

We would like to thank to the Reviewer for his/her comments especially about the presentation of data in figures.

Ad additional experiments: Number of animals was determined in advance according to previous experience with the given models and corresponded with the number of animals in the Project of Experiment approved by the Central Committee for Animal Protection. To increase the number of animals to reach the level of statistical significance is highly questionable and it is not in a line with the 3Rs principle.

Figures were changed. There is a new set of figures demonstrating box plots for PTZ-elicited seizures and lines for cortical afterdischarges. Text to figures was carefully prepared taking into account all comments.

Ll.33-36 Glutamate-related drugs used for treating epilepsy in humans: Following sentence was added (now ll.35-38): Some antiepileptic drugs affecting glutamate ionotropic receptors are used in human medicine – e.g. felbamate (among other mechanisms of action antagonist of NMDA receptors – Pellock and Brodie 1997), perampanel (AMPA receptor antagonist – Hanada 2914).

Ll.115-127:

When PEAQX was administered: It was administered 30 min before PTZ (now ll.111-113).

How many animals received each dose of PEAQX: All dose groups in the PTZ model consisted from 8 rats in P12 and P25, ten animals in P18. In the cortical epileptic afterdischarges part of experiments, majority of groups was also composed from 8 rats but exception was formed by the 12-day-old 10- and 20-mg/kg group and 18-day-old 5- and 20-mg/kg groups with only six animals. The numbers are now presented in Text to Figures.

Experiments were double-blind: Statement is in 2.5. Statistics, l.174: Data acquisition and analysis were done blinded to the treatment.

There were no control group?: Control groups: In addition to clearly marked controls in all graphs we included following information also into the text (2.2.Drugs, ll.109-111): In all three age groups in both experiments animals in the control groups received physiological saline in the volume corresponding to the highest dose of PEAQX (i.e. 4 ml/kg s.c.).

Figures: Figures were changed – see above.

Remark to ll.245-250: Comparisons with control groups was made in both models as presented in the para 2.5. In addition, two-way RM ANOVA was used for evauation of cortical afterdischarges (ll.181-185): Two-way ANOVA RM with one between-group factor (control, treatment or age) and one within-subject factor (repeated session), corrected for multiple comparison by controlling the FDR was used to compare duration of epileptic afterdischarges between treatment groups within one age group and/or between control groups of various ages.  We also changed the text to Figures.

LL.245-250 The statistical results are now in the text.

Reviewer 2 Report

This paper by Mares P and Kubova H. demonstrates interesting findings concerning the developmental profile of the anticonvulsant action of PEAQX in immature rats subject to the three induced models of epileptic seizures. PEAQX is an antagonist of NMDA receptors that subunits composition depends on developmental stages and the brain regions.

  Epilepsy is the fourth most common neurological disorder very often with an unknown cause. Although there are available on market anti-epileptic drugs (AEDs), that helps control seizures, a side effect of long term administration, especially in paediatrics is a major concern. The following paper enters precisely with needs for looking of new age-specific therapies. My general comment is such that it would be worth to include in the future experimental paradigm different strains of rats and conducted them on both sexes, despite the fact that in tested age, the rats are sexually immature. The authors are aware of the complexity of the tested process which is reflected in the interpretation of the results. I consider being evidence of the maturity of the scientific approach. Summarizing the presented in the paper findings are worth publishing, however, minor changes need to be addressed:

  1. Line 55: It is worth updating the list of publications with the latest items.
  2. Line 58: This is the multiple complex sentence, it is worth simplifiing it.
  3. Line 118: I would suggest to introduce abbreviation (mMS) after “seizures”.
  4. General comment to all figures: I’d suggest present the plots as scatter plot with bars; in the drawing caption add information about the N (numbers of animals), add explanation concerning error bars,
  5. Figure 2: unify format of a dot between the plot and caption; please clarify the meaning of n in the middle graphs in the caption.
  6. Figure 3: add information about the N - numbers of animals among the experimental group treated with a particular dose of PEAQX in the caption

Author Response

Reviewer 2

Many thanks for your detailed comments. They helped a lot to improve the ms.

  1. Line 55: A review article was added into this line
  2. Line 58: The sentence (now on ll.61-63) was divided in two: A non-competitive antagonist of NMDA receptors MK-801 (dizocilpine) possesses marked anticonvulsant effects. They have been demonstrated in several models of epileptic seizures across all age groups of laboratory rodents [12,13].
  3. Abbreviation (mS for minimal clonic seizures) was introduced and used throughout the ms.
  4. Figures were changed, they are now presented as scatter plot with bars, Captions were changed according to the new format of figures.
  5. The symbols were checked, there are now only circles, not dots in figures with cortical epileptic afterdischarges.

Numbers in the figures 1A-C (e.g. n=1 or n=2) means that one or two rats exhibited the phenomenon and therefore mean and measure of variability are not presented. Explanation of these numbers is now in the text to figures.

  1. Number of animals is now presented in text to figures.

Reviewer 3 Report

The paper Mares and Kubova explores the age and dose-dependent effects of inhibitor of glu2A subunit of NMDA receptors PEAQX in 12-, 18- and 25-day-old rats in three models of seizures: PTZ- induced seizures originating primarily in the brainstem, (2) in the forebrain and (3) cortical epileptic afterdischarges generated in the cortico-thalamo-cortical circuits. The manuscript is well written and the findings are of considerable interest. Specific comments are listed below

Title

The title is not correct. The immature rodents is the animals younger than 21 postnatal days, I suggest to use “juvenile rats”

Abstract

Line 12 -16. The sentence must split up according to the models of seizures.

It is necessary to describe in more detail the data of two models of PTZ-induced seizure

Line 19-20 The statement about the role of Glu2B subunit of NMDA receptors is speculative.

Methods

The origin of the recording equipment is necessary. Authors should analyze not only duration of cortical afterdischarge but also a spectral power of seizures. Authors should present the original electrophysiological data in the results section.

Results

In general, the author should explain why in PTZ induced seizures were analyzed only by behavior characteristics and for cortical model of seizure was used electrophysiology recording without presentation the motor correlates. EEG recordings in the basal forebrain and in the brainstem during PTZ administration could reveal the origin of seizures in this model that make the conclusion more relevant.

  1. I strong recommend to show and analyze EEG recodings (cortical epileptic afterdischarges) because electrographic properties of epileptic afterdischarges depend on the brain region stimulated. Spike-and-wave patterns can be elicited by AD induction either in the neocortex or mediodorsal thalamus. In contrast, AD induction in limbic circuits produces fast spikes, large delta waves, and/or sharp theta waves (Kandratavicius et al 2014).
  2. Also I suggest to use two way ANOVA for comparison between age animals and number of stimulus or dose of PEAQX
  3. Page 7 line 239 “PEAQX had no effect on severity of motor correlates accompanying ADs”  - please provide the analysis of motor correlates  and explain the absence of PEAQX effects on clonic seizures in this model

Figures

  1. Figure 2 (incident of generalized seizures ) at 12 and 25 days – no asterisk at high dose of PEAQX
  2. Figure 2: Dots indicated in the figures do not correlate with the description in the legend (open circle).
  3. Figure 3: the color code of groups during pre-drug stimulation is misleading because at this stage (â„–1) there is no injection of PEAQX
  4. Figure 3 should be described in more detail in the legend especially concerning the significance labels

Discussion

  1.  Authors should discuss the possible involvement of GABA and AMPA receptors in the seizures especially for immature rats as PTZ is a selective antagonist of receptor of GABAA (Ueda et al Neurochem Res 2009) In a related study it was reported that AMPA receptors play role in cortex and basal ganglia for sustaining PTZ kindling phenomenon (Ekonomou et al Brain Res Mol Brain Res 2001)
  2. Page 9 line 266 : Citation of paper Neyton an Paoletti in this sentence “PTZ–induced generalized tonic-clonic seizures are primarily generated in the brainstem” is not correct
  3.  In my opinion, the division of PTZ-induced seizure by source of generation is relatively poor. There are only several papers (Browning 1985; Browning, R.A., Nelson. 1986, Gale 1988) where the brainstem as a generator of generalized tonic-clonic seizures was hypothetically suggested without clear argumentation. Are there any other literature or your own data supporting the localization of GTCS in the brain stem after PTZ injection
  4. Is it possible that subunit composition of NMDA channels and/or or sensitivity to GLUN2 inhibitors  change during epilepsy. For example the epileptic seizures may alter the functional properties of NMDARs or redistributed of receptors of different subunit composition between extrasynaptic and synaptic sites after seizures. In a lithium-pilocarpine model of epilepsy, SE led to the fast relocation of obligate GluN1 subunits and an increasing number of NMDARs presynapse (Naylor et al. 2013). The incorporation of already existing NMDARs into the synapses may be accompanied by alterations in gene expression of different NMDAR subunits. Stronger inhibition of NMDAR-mediated responses by ifenprodil in post-SE animals suggested that the proportion of GluN2B-containing NMDARs increased in an SE model (Postnikova et al 2019)Amakhin et al. 2017; Naylor et al. 2013). The increased localization of the GluN2B subunit in extrasynaptic and presynaptic sites together with a concomitant decrease at postsynaptic compartments was reported in epileptic tissue (Frasca et al. 2011).

References

The order of references in the text is wrong. The paper by 40 was citied early than 39  (page 9 and 10)

Author Response

Reviewer 3

I would like to thank you for all comments and especially for the comment on EEG recordings. This comment not only led to changes in this ms but also to modification of our plans for future experiments.

As analysis of EEG recordings concerns, our young collaborator Grigoriy Tsenov, PhD, adapted the program, prepared and evaluated the examples of results and participated in writing a final version of the ms. Therefore he is now included as a coauthor of our ms.

To analyze all recordings could take a few months, therefore only examples were prepared. The thorough analysis will remain to the next study. I am deeply interested in postictal period and the analysis of pre- and postictal EEG will form a part in the study already started (with evoked potentials).

Title

The expression „immature“ (which is common in developmental studies) was replaced by „developing“

Abstract

The sentence on the lines 12-16 was split according to models of seizures and description of generalized seizures elicited by pentylenetatrazol was completed with an important phenomenon „with a loss of righting reflexes“.

Ll.19-20: The possibility that the effect of the 20-mg/kg dose in the two younger groups might be an effect of GluN2B receptors due to low selectivity of PEAQX is hypothetical possibility which might explain this finding. Formulation in the abstract was a little changed to express this hypothesis: „The highest dose (20 mg/kg) was efficient also in the two younger groups what might be due to low selectivity of PEAQX.“

Methods

Spectral analysis was performed and examples are presented in the text. Detailed analysis of pre- and postseizure EEG as well as EEG pattern of afterdischarges will form a part of a study of postictal changes.

Results

Motor correlates of cortical stimulation and epileptic afterdischarges are now described: Cortical stimulation and afterdischarges elicited clonic movements (seizures) of forerlimbs (Racine stage 3). Exception was formed by one or two animals in 18- and 25-day-old groups, where stage 4 (forelimb clonuc + rearing) was observed. Axerage stage never exceed the value 3.25±0.164, ANOVA did not indicate significant differences. 

Subcortical recordings are difficult to perform in animals younger than 2 week, because their skull is not fully ossified and it is still somehow “elastic”. For this reason, it is very difficult to fix registration electrodes firmly enough to the skull. Thus, during severe convulsions (GTCS) movements of deep electrodes are source of numerous artefacts in the EEG recordings. Therefore EEG registration from the brainstem is not regularly used in screening experiments.

As concerns your questions:

  1. EEG recordings are demonstrated in Results p.12, ll.297-301 and in Figure 2. Results of analysis are in Fig.4. We used only cortical stimulation, stimulation electrodes were over sensorimotor cortical area (ll.138-139).
  2. Two way ANOVA was used and present in the revised version of ms.
  3. Motor correlates of ADs: Semiology of motor phenomena accompanying cortical epileptic afterdischarges is practically identical with minimal clonic seizures elicited by PTZ and many other convulsant drugs. Details on motor clonic seizures were added into the text. We do not have an explanation for the resistance of minimal clonic seizures but we found similar phenomenon in studies on many other drugs (data published during the last 30 years).

Figures (remarks 1-4)

Figures were changed and their description carefully formulated.

Discussion

  1. Possible involvement of GABA and AMPA receptors: GABA and AMPA receptors surely play a role in generation of seizures but to discuss this problem in our ms will bring at least two more pages. In addition, other receptors are involved – e.g.glycine and kainic acid. The role of GABA in cortical epileptic afterdischarges is surely not principal (Tabashidze and Mareš 2011, Mareš et al. 2020).
  2. Citation of Neyton and Paoletti: Thank you for finding this mistake. Instead of citations of Neyton and Paoletti there are now citations [21,22] and in the next line about the corticothalamocortical phenomenon are [23,36]. The correction was made.
  3. Papers of Browning demonstrate that minimal clonic seizures cannot be elicited in rats with transection of the brain stem (e.g.midcollicular transection). In contrast, generalized tonic-clonic seizures are present in the animals with transection. I checked these data and my results agree with Browning’s results – rats with transection of the brainstem are not able to generate minimal clonic seizures. Generalized tonic-clonic seizures were regularly elicited in the transected animals. Spinal cord transection at the low thoracic level did not block the possibility to evoke tonic-clonic seizures in hindlimbs, only the threshold was higher than for the same phenomenon in forelimbs. These results were added to the ms as citation [22] and subsequent citations were renumbered.
  4. Possible changes of subunit composition of NMDA channels: We are fully aware of the changes of NMDA receptors in postSE model of epilepsy. Data in our manuscript are valid for epileptic seizures (not epilepsy with generation of spontaneous seizures as in the postSE model) and for short time – the sixth cortical epileptic afterdischarge was elicited 100 minutes after the first one.

References

One more citation on the site of origin of two different types of PTZ-induced seizures was added (Mareš 2006) and the numbering and order of references were corrected. I have to apologize for a mistake in the original ms.

Round 2

Reviewer 1 Report

I find the authors nearly solved all the criticism raised by reviewers

Reviewer 3 Report

The authors have responded to all my comments, the revised version of the manuscript appears to be good.